# Multiple sclerosis patients have an altered gut mycobiome and increased fungal to bacterial richness

Meeta Yadav[1,2☺], Soham Ali[3☺], Rachel L. Shrode[4], Shailesh K. Shahi[1], Samantha N. Jensen[1,5], Jemmie Hoang[6], Samuel Cassidy[7], Heena Olalde[7], Natalya Guseva[1], Mishelle Paullus[7], Catherine Cherwin[6], Kai Wang[8], Tracey Cho[7], John Kamholz[7], Ashutosh K. Mangalam[1,4,5,9]*

1 Department of Pathology, Carver College of Medicine, University of Iowa, Iowa City, IA, United States of America, 2 University of Iowa College of Dentistry, Iowa City, IA, United States of America, 3 Carver College of Medicine, University of Iowa, Iowa City, IA, United States of America, 4 Informatics Graduate Program, University of Iowa, Iowa City, IA, United States of America, 5 Interdisciplinary Graduate Program in Immunology, University of Iowa, Iowa City, IA, United States of America, 6 College of Nursing University of Iowa, Iowa City, IA, United States of America, 7 Department of Neurology, University of Iowa Hospitals and Clinics, Iowa City, IA, United States of America, 8 Department of Biostatistics, College of Public Health, University of Iowa, Iowa City, IA, United States of America, 9 Iowa City VA Health System, Iowa City, IA, United States of America

☺ These authors contributed equally to this work.
* Ashutosh-mangalam@uiowa.edu

**Data Availability Statement:** The microbiota 16S and ITS2 sequencing analysis data have been deposited to the Sequence Read Archive (https://www.ncbi.nlm.nih.gov/sra) under PRJNA732670

## Abstract

Trillions of microbes such as bacteria, fungi, and viruses exist in the healthy human gut microbiome. Although gut bacterial dysbiosis has been extensively studied in multiple sclerosis (MS), the significance of the fungal microbiome (mycobiome) is an understudied and neglected part of the intestinal microbiome in MS. The aim of this study was to characterize the gut mycobiome of patients with relapsing-remitting multiple sclerosis (RRMS), compare it to healthy controls, and examine its association with changes in the bacterial microbiome. We characterized and compared the mycobiome of 20 RRMS patients and 33 healthy controls (HC) using Internal Transcribed Spacer 2 (ITS2) and compared mycobiome interactions with the bacterial microbiome using 16S rRNA sequencing. Our results demonstrate an altered mycobiome in RRMS patients compared with HC. RRMS patients showed an increased abundance of *Basidiomycota* and decreased *Ascomycota* at the phylum level with an increased abundance of *Candida* and *Epicoccum* genera along with a decreased abundance of *Saccharomyces* compared to HC. We also observed an increased ITS2/16S ratio, altered fungal and bacterial associations, and altered fungal functional profiles in MS patients compared to HC. This study demonstrates that RRMS patients had a distinct mycobiome with associated changes in the bacterial microbiome compared to HC. There is an increased fungal to bacterial ratio as well as more diverse fungal-bacterial interactions in RRMS patients compared to HC. Our study is the first step towards future studies in delineating the mechanisms through which the fungal microbiome can influence MS disease.

for free public access. All metadata for sequencing files are provided in S1 Table.

**Funding:** This work was supported by a grant from Schwab Foundation Margaret Heppelmann and Michael Wacek, Departmental Startup fund, NIH R01 grant 1R01AI137075-01, and VA Merit Award 1I01CX002212 to Ashutosh Mangalam. Meeta Yadav is supported by NRSA T90 training grant 5T90DE023520. Rachel Shrode is funded by the University of Iowa's Informatics Fellowship from the Informatics Graduate Program. Soham Ali is supported by the Emory Warner Medical Student Research Fellowship from the University of Iowa Department of Pathology.

**Competing interests:** AM is one of the inventors of a technology claiming the use of Prevotella histicola to treat autoimmune diseases. AM received royalties from Mayo Clinic (paid by Evelo Biosciences). MY, SA, RS, SS, SJ, JH, SC, HO, NG, MP, CC, KW, TC, JK declare no commercial or financial relationships that could be a potential conflict of interest.

## Introduction

Multiple Sclerosis (MS) is a neuroinflammatory autoimmune disease that affects ~2.5 million people worldwide. Though MS has different clinical subtypes, most of the patients (~85%) present with relapsing-remitting MS (RRMS). The precise etiopathogenesis of MS is unclear, but both genetic and environmental factors have been suggested to play an important role in susceptibility and the pathogenesis of MS. In recent years multiple groups, including ours, have highlighted the role of gut bacteria in the pathobiology of MS [1–8], and it has emerged as an important environmental factor. However, the gut microbiome consists of non-bacterial microbes such as fungi, viruses, and archaea [9,10], yet the role of the fungal microbiome (mycobiome) is not well studied in MS.

Although fungi make up approximately 0.1% of the gastrointestinal tract [10,11], their impact on human health is significant due to their ability to regulate local and systemic host immune responses [12,13]. The importance of the fungal microbiome on the immune system was demonstrated in a study where mice that were rewilded had increased intestinal abundance of fungi along with an expansion of granulocytes compared to laboratory mouse controls [13]. While in human peripheral blood 50–70% cells are neutrophils, only 10–25% cells are neutrophils in mouse peripheral blood [14]. Additionally, an oral antifungal treatment of mice in a colitis model showed increased disease severity and exacerbated allergic airway disease with increased *Aspergillus*, *Wallemia*, and *Epiccocum* and decreased *Candia* spp [12]. Moreover, oral administration of a mixture of these three fungi (*Aspergillus amstelodami*, *Epicoccum nigrum*, *and Wallemia sebi*) was sufficient to recapitulate the exacerbating effects of antifungal drugs on allergic airway disease [12]. The importance of the mycobiome in human health and disease is validated by studies showing alterations in mycobiome composition in various autoimmune diseases such as inflammatory bowel disorders (IBD), ankylosing spondylitis (AS), and type 2 diabetes mellitus (DM) and in pwMS (people with MS) compared to healthy controls (HC) [15–18]. These studies have also shown correlations between bacterial and fungal microbiota that are altered in disease states. The relationship between gut bacteria and fungi has been demonstrated in the murine model where bacteria outnumbered fungi by greater than three times prior to antibiotic administration, but after antibiotic administration bacterial abundance dropped three-fold while fungal abundance increased around 40-fold [19]. Thus, all these observations suggest that the gut mycobiome might play an important role in the health and disease including MS.

Based on the association of gut mycobiota in other autoimmune diseases and also in pwMS, which was recently showed by Shah et al [18], we hypothesized that the gut fungal microbiota might be altered in MS patients. Therefore, in this study we profiled the fungal microbiota and associated fungal functional characteristics in RRMS patients (MS) (n = 20) and compared them with those of HC (n = 33). We also analyzed the bacterial microbiome of the same patients to determine the correlation between the fungal and bacterial microbiome. We observed that the MS patients have a distinct fungal microbiome compared to HC, with differential abundances of multiple fungal genera and fungal functions as well as altered fungal-bacterial relationships. A distinct fungal microbiome in RRMS patients could have implications in the pathogenesis of MS, potentially through functional differences and altered interactions with the bacterial microbiome.

## Methods

### Standard protocol approvals, registrations, and patient consents

The study was done in accordance with the guidelines approved by the University of Iowa Institutional Review Board (IRB) (#201512717). A prior written informed consent was obtained from all the subjects to participate in the study.

**Table 1. Biometric and MS patient treatment data.**

|  | HC | MS | p-value |
|---|---|---|---|
| **Age: Mean ± SD** | 42 ± 14 | 43 ± 7.7 | t-test: 0.61 |
| **Sex: Male/Female** | 5 / 28 | 5 / 15 | Fisher: 0.48 |
| **BMI: Mean ± SD** | 24 ± 3.7 | 30 ± 7.9 | t-test: 0.015 |
| **Treatment: Yes / No** | NA | 16 / 4 | NA |
| **Treatment** |  |  |  |
| Interferon beta |  | 3 |  |
| Glatiramer acetate |  | 4 |  |
| Ocrelizumab |  | 3 |  |
| Dimethyl fumarate |  | 6 |  |
| Not on treatment |  | 4 |  |

## Human subjects enrollment

Relapsing-Remitting Multiple Sclerosis (RRMS) patients (n = 22) who fulfill the McDonald diagnostic criteria for MS were recruited from the Neuroimmunology Clinic at the University of Iowa Hospital Clinics (UIHC) at the University of Iowa. Healthy controls (HC) (n = 34) were also recruited through the University of Iowa (Table 1). Individuals eligible for the study were those 18–63 years of age with a diagnosis of RRMS. All patients were either on MS treatment or not on treatment for at least three months prior to enrollment. Patients and healthy controls were asked to provide samples at least four weeks after their last dose of oral antibiotics or laxatives, and at least three months after their last colonoscopy. Those with active pregnancy or a history of bariatric surgery were excluded from the study.

## Human sample collection

Stool samples were collected by patients and HC in Commode Specimen Collection kits (Fisher PA, USA) provided to them by our laboratory along with the instruction sheet and shipped on frozen gel packs to our laboratory by overnight delivery. Once in the laboratory, the stool was aliquoted and stored at -80 degrees until DNA extraction. The same collection kits, instruction sheets, and processing methods were used for fecal samples from MS patients and HC.

## DNA extraction and sequencing

Microbial DNA was extracted from each fecal sample using Qiagen DNeasy PowerLyser PowerSoil Kit (Qiagen, Germantown, MD) per the manufacturer's instructions using a bead-beating step (PowerLyzer 24 Homogenizer, Omni International, USA). The fungal microbiome was analyzed using Two-step PCR as described previously for bacterial microbiome analysis [20] except instead of 16S rRNA, we targeted the internal transcribed spacer 2 (ITS2) region of fungal 18S rRNA using the following primers: forward primer 86 F(5′-3′) GTGAAT CATCGAATCTTTGAA and reverse primer 4R (5′-3′) TCCTCCGCTTATTGATATGC [21]. We choose ITS-2 primer over others (ITS-1, 18s- etc.) as a previous study comparing different fungal primers showed that ITS-2 specific primer allowed detection of maximum number of fungal OTUs [21]. Fungal PCR conditions were as follows: 95°C for 5 min, 35 cycles of 95°C for 30s, 59°C for 30s, 72°C for 30s, and 72°C for 10 min. Bacterial 16S rRNA sequencing was performed as per protocol published by our lab [20].

## Metagenomic profiling

The raw fungal ITS2 (internal transcribed spacer 2) sequence data was processed utilizing DADA2 [22]. In brief, the reads were trimmed to remove the primer sequences, then truncated based on a quality score of 25. Reads were then denoised to infer the exact sequence variants by resolving single-nucleotide errors. Paired forward and reverse reads were merged, and any resulting chimeras were removed to produce amplicon sequence variants (ASVs). These ASVs were then assigned fungal taxonomy at the genus level using a naive Bayesian classifier method and the UNITE database (Version 8.2) [23]. After the removal of three samples (2 MS and one HC) with less than 1000 reads, 20 MS samples and 33 HC samples were available for fungal profiling, with a median number of reads of 37,331.

The raw sequence data of the V3-V4 region of bacterial 16S rRNA was processed utilizing DADA2. These sequences were processed as described above for ITS sequences, and the final ASVs were assigned bacterial taxonomy at the kingdom to species levels using the Silva database (version 138.1) [24] with a median number of reads of 49,976.

## Functional profiling

The functional profile of the gut mycobiome was generated using FungalTraits database [25]. Of the 142 identified fungal genera in this study, 86 had functional entries in this database, and the functional profile of each sample was then inferred using its fungal composition and the functional data of each of these 86 fungal genera.

## Statistical analysis

Analysis and figure generation of alpha diversity, beta diversity, and differential abundance of all features was performed entirely in R (Version 4.0.3) using the vegan [26] and ggpubr [27] packages with custom R scripts. Prior to statistical analysis, the fungal and bacterial datasets were each constant-sum scaled to one million reads and generalized log-transformed. The fungal and bacterial data were then filtered to remove low prevalence taxa. Alpha diversity and differential abundance analysis between the MS and HC groups were performed using the Wilcoxon test, and p-values were adjusted using the Benjamini-Hochberg algorithm. PERMANOVA was performed to analyze the statistical significance of beta-diversity clustering using the adonis2 function from the vegan package.

Inter-kingdom correlation analysis was conducted between the bacteria and fungi at the genus level using Spearman's rank correlation, and only correlations showing p-value less than 0.05 are reported. Bootstrapped random forest analysis was conducted using the Boruta package [28] at the suggested significance level of 0.01 with 500 trees for 100 iterations.

# Results

## Fungal microbiota of RRMS patients is different from healthy controls

To characterize the fungal microbiome diversity, we sequenced fecal samples of 20 RRMS patients and 33 healthy controls (HC) using ITS2 rRNA sequencing. A total of 136 fungal genera were identified, with 31 unique to the HC group, 73 unique to the MS group, and 32 present in both groups. Alpha diversity using the Chao1 index showed greater richness in MS than in HC (Fig 1A), though this was not statistically significant ($p > 0.08$). Principal coordinate analysis of beta diversity using Bray-Curtis dissimilarity (Fig 1B) demonstrated distinct clustering of the mycobiome of MS compared with HC ($p = 0.011$), while gender ($p = 0.32$) and Body Mass Index (BMI) status ($p = 0.73$) did not show a significant effect on mycobiome composition. Interestingly, beta-diversity analysis of the MS group did not demonstrate a

significant effect of treatment with dimethyl fumarate (p = 0.872) on the mycobiome composition (S1A Fig). Including other disease modifying treatments (DMTs) with at least 3 samples, no significant effect was shown for treatment (p = 0.163; S1B Fig), gender (p = 0.197), or BMI status (p = 0.414).

Overall, the two most prominent phyla by relative abundance were *Ascomycota* and *Basidiomycota* (Fig 1C), and the top five fungal genera were *Saccharomyces*, *Candida*, *Malassezia*, *Penicillium*, and *Cladosporium* (Fig 1D), comprising 75.5% of all the identified fungal genera in the samples.

Examining the differential abundance between the MS and HC groups at the phylum level showed a lower abundance of *Ascomycota* (padj = 0.011) and higher abundance of *Basidiomycota* (padj = 0.011) in the MS group compared to HC (padj = 0.011 and 0.011, respectively). Previous studies in IBD and AS have shown that there is a strong negative correlation between *Basidiomycota* and *Ascomycota* characterized by a higher *Basidiomycota*/*Ascomycota* ratio in IBD [17] and AS [16]. In this study, we also observed that MS patients have an increased *Basidiomycota*/*Ascomycota* ratio (p = 0.0053) compared to HC (Fig 1E). Thus, our data demonstrates a phylum-level shift in MS towards *Basidiomycota* and away from *Ascomycota*. Interestingly, within the *Ascomycota* phylum, the genera *Candida* (padj = 0.021) and *Epicoccum* (padj = 0.021) were more abundant in the MS group (Fig 2A & 2B), while only *Saccharomyces* (padj = 0.021) was depleted in the MS group (Fig 2B) compared with HC. Species-level analysis showed *Saccharomyces cerevisiae* comprising 88% of the *Saccharomyces* genus and *Candida albicans* comprising 81% of the *Candida* genus across all samples. Thus, our study finds that MS patients have a distinct mycobiome compared to HC with enrichment or depletion of certain fungi.

Random **forest analysis** to identify the potential discriminating fungi. We next assessed the ability of the gut mycobiota profile to predict disease phenotype in our samples. A bootstrapped random forest algorithm of 500 trees was used to generate a predictive model based on the gut mycobiota profiles of the samples. The Boruta algorithm was then utilized to identify the fungal genera that were most important in distinguishing between MS and HC samples at a significance level of 0.01.

Interestingly, the four aforementioned genera, *Candida*, *Saccharomyces*, *Epicoccum*, and *Malassezia*, were again identified (Fig 2C). The Boruta algorithm additionally identified *Penicillium* which is decreased in MS, and *Malassezia*, which is increased in MS (Fig 2C), as a significantly important feature in distinguishing MS from HC. Thus, the random forest classification analysis suggests that these fungal taxa could be important discriminators between MS and HC.

## Functional profile of gut mycobiome in MS patients

Analysis of the mycobiome functional profile determined from the FungalTraits database revealed several enzymes whose enrichment differed significantly between MS and HC samples. Levels of amino acid permease (padj = 0.029), cellobiohydrolase (padj = 0.011), endoglucanase (padj = 0.029), and invertase (padj = 0.0211) were lower in MS patients compared to HC (Fig 3), while amylase (padj = 0.011) was increased in these patients (Fig 3). These results suggest potential differences in fungal metabolism and functional phenotypes in the gut of MS patients.

Gut bacterial microbiome of RRMS patients is different from healthy controls. We also characterized the bacterial microbiome of the same samples using 16S rRNA (V3-V4) sequencing. A total of 285 bacterial genera were identified, with 196 genera in the HC group and 185 genera in the MS group. Alpha diversity by Chao1 index demonstrated decreased

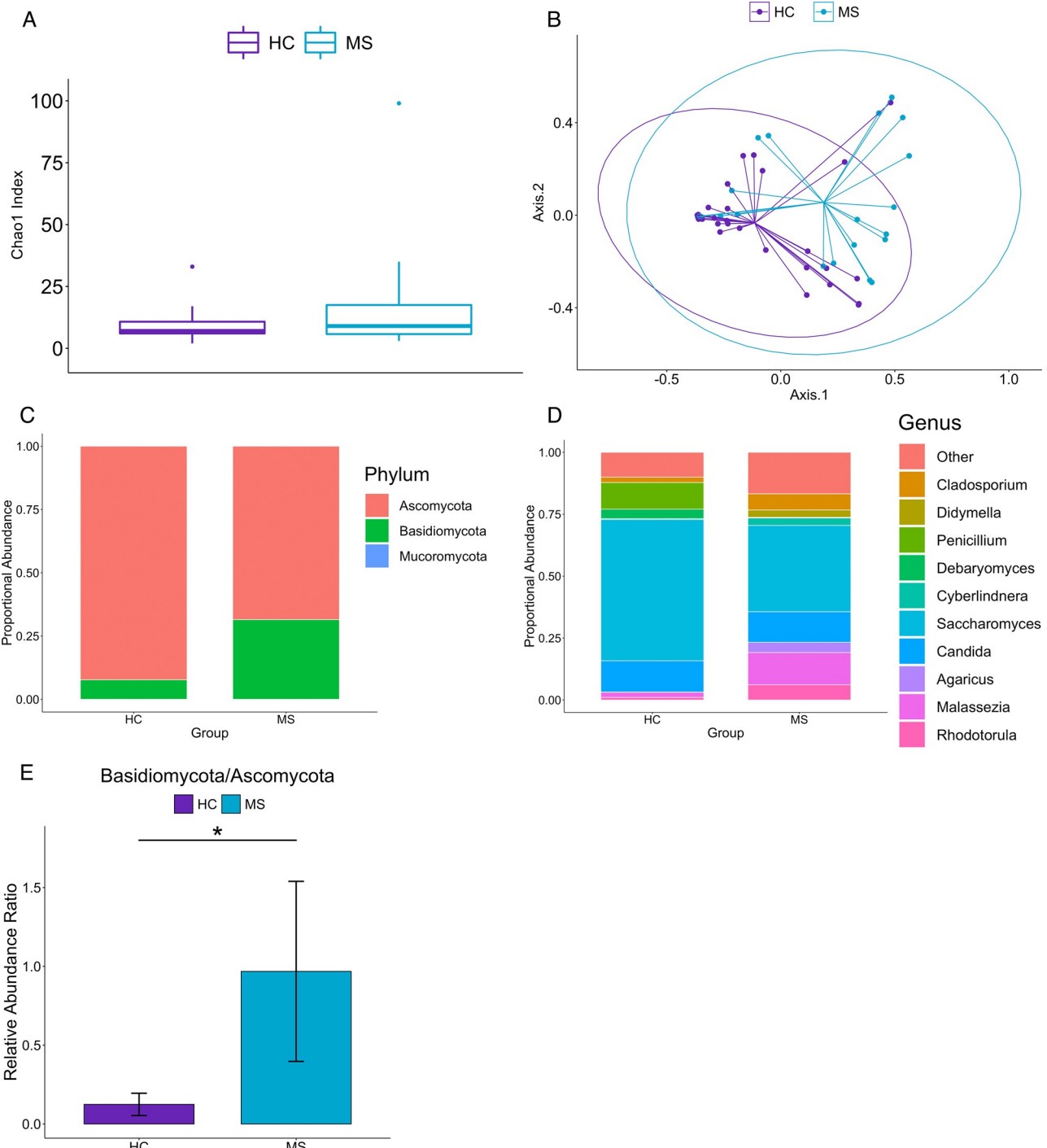

**Fig 1. Fungal microbiota of RRMS patients is different from healthy controls.** (A) Fungal ASV richness estimated by Chao1 index in MS and HC. (B) Principal coordinate analysis of Bray-Curtis dissimilarity of HC and MS shows that the mycobiome of HC and MS are distinct (PERMANOVA: p = 0.011). Ellipses correspond to a 95% confidence intervals around the centroids for each group. (C) Bar plot showing the relative abundances of fungal phyla. *Basidiomycota* was increased and *Ascomycota* was decreased in MS compared to HC. (D) Bar plot showing the top 10 fungal genera in HC and MS (determined by average relative abundance across all samples). The top 10 genera account for 85.5% of all identified fungal genera. (E) *Basidiomycota/Ascomycota* ratio is significantly increased in MS (p = 0.0053).

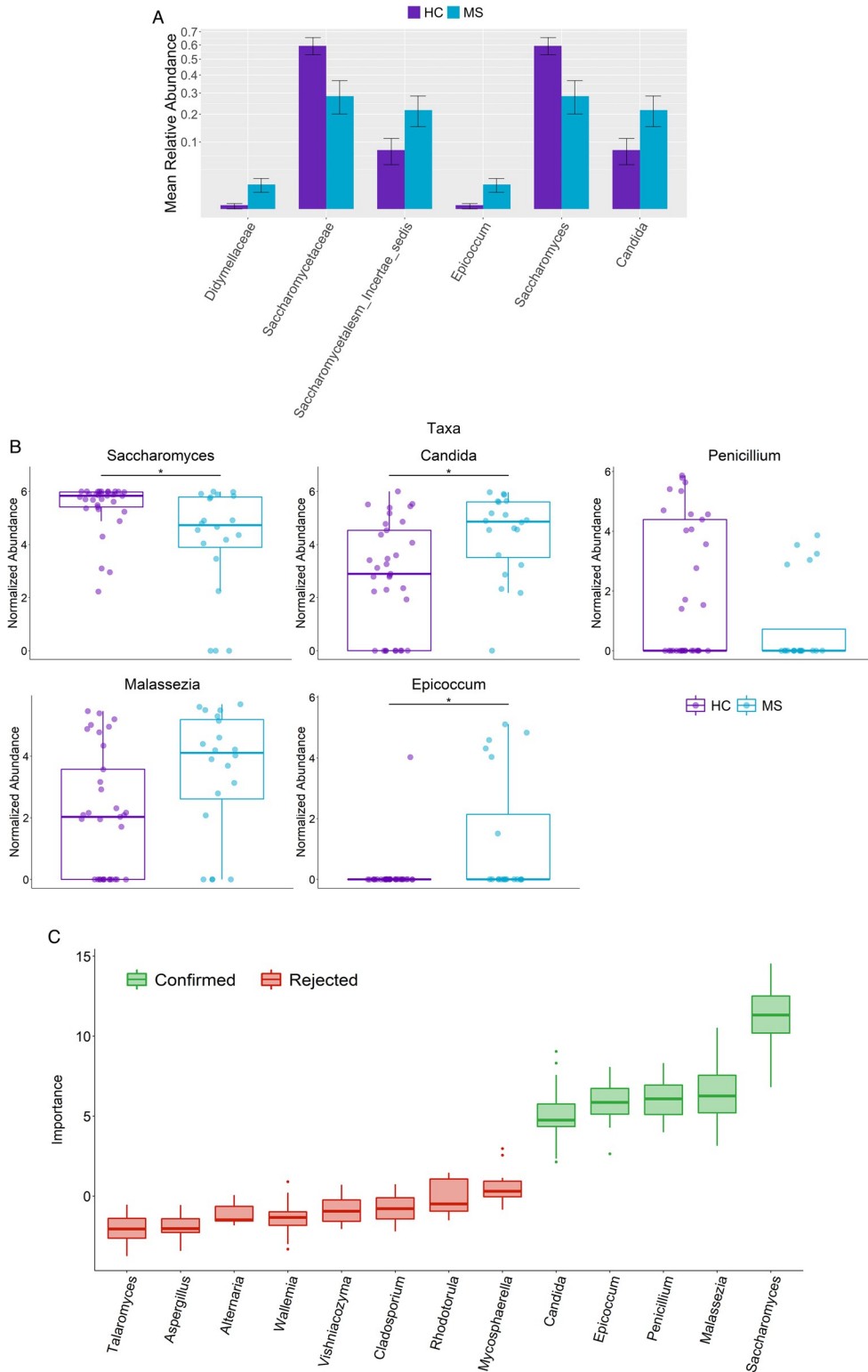

**Fig 2. Differentially abundant fungal genera in RRMS versus healthy controls.** (A) Bar plot showing relative abundances of differentially abundant taxa (p < 0.05) at the phylum, family, and genus level. (B) Differentially abundant fungal genera in MS and HC using Wilcoxon signed rank test and adjusted for multiple comparisons with the Benjamini-Hochberg method at a significance level of 0.05. *Candida*, *Epicoccum*, and *Malassezia* are increased in MS compared to HC. *Saccharomyces* is decreased in MS compared to HC. *Penicillium* was identified in random forest

analysis as a significant feature and was decreased in MS, though the Wilcoxon test did not reach statistical significance after adjusting for multiple comparisons. Abundance values are sum-scaled to 1 million and generalized log-transformed. The * symbol indicates p-value <0.05. (C) Importance of features determined by random forest and tested for significance with the Boruta algorithm at a significance level of 0.01.

bacterial richness in the MS group compared with the HC group (p = 0.020; Fig 4A), aligning with our previous study showing less bacterial diversity in the gut of MS patients [2]. Additionally, the MS and HC groups clustered separately on principal coordinate analysis of beta diversity using Bray-Curtis dissimilarity (p = 0.004; Fig 4B), indicating distinct microbiota profiles.

The five most prominent bacterial phyla across all the samples were *Actinobacteria*, *Bacteroidetes*, *Firmicutes*, *Proteobacteria*, and *Verrucomicrobiota*, accounting for 98.8% of all identified bacteria (S2 Fig), and of these the top 50 genera account for 94.8% of all identified bacteria (Fig 4C) with *Bacteroides*, *Alistipes*, *Agathobacter*, *Blautia*, *Faecalibacterium*, and *Akkermansia* comprising around 50% of all genera.

Within the *Bacteroidetes* phylum, the *Barnesiellaceae* (padj = 4.4e-3) family and one of its genera, *Barnesiella* (padj = 5.9e-3), as well as the genus *Odoribacter* (*Odoribacteraceae* family; padj = 0.045) were decreased in the MS group compared to HC (S3 Fig and Fig 4D). The other differentially abundant family, *Eggerthellaceae* (*Actinobacteria* phylum; padj = 6.3e-3) and one of its genera, *Eggerthella* (padj = 2.4e-4), were overabundant in MS compared to HC (S3 Fig and Fig 4E). Within the *Firmicutes* phylum, an uncultured genus of the *Oscillospiraceae* family (*Oscillospiraceae* UCG 003) was depleted in the MS group compared to HC (padj = 0.045) while *Blautia* (padj = 0.045) and *Hungatella* (padj = 0.045) were increased in MS (Fig 4E). We have previously also shown *Prevotella* to be significantly decreased in MS patients and beneficial in alleviating symptoms of EAE in mice through modulation of the gut microbiome [2,29]. In this study, the relative abundance of *Prevotella* was decreased in the MS group, though not significantly (padj = 0.70; Fig 4C). Similarly, *Akkermansia* was increased in MS but did not

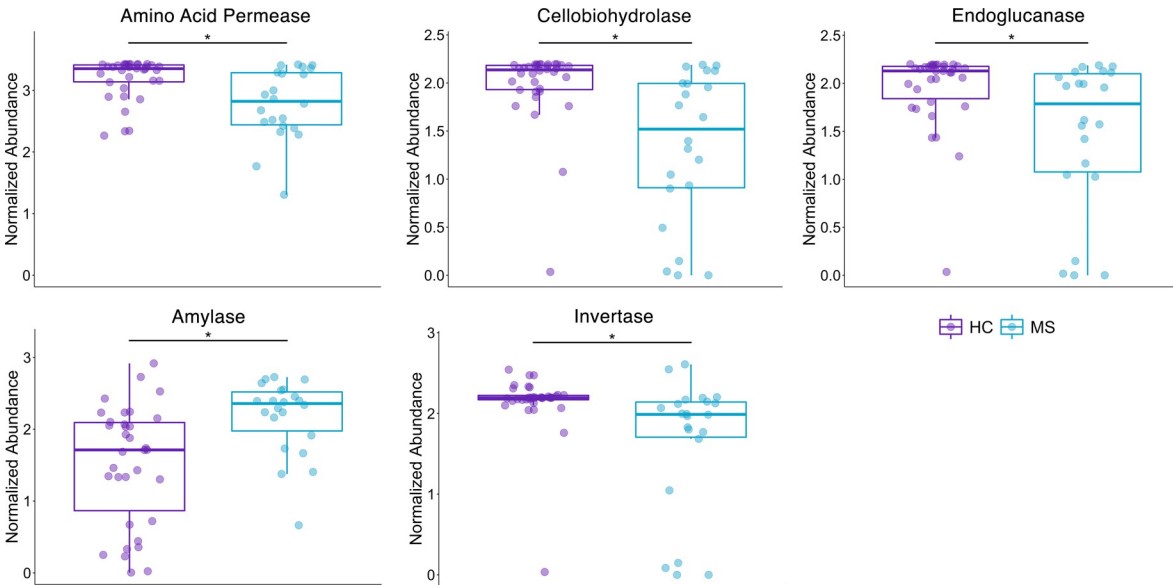

**Fig 3. Functional profile of gut mycobiome in MS patients.** Differentially enriched fungal functions using Wilcoxon signed rank test and adjusted for multiple comparisons with the Benjamini-Hochberg method at a significance level of 0.05. Amino acid permease, cellobiohydrolase, endoglucanase, and invertase are decreased in MS compared to HC. Amylase is increased in MS compared to HC. Abundance values are sum-scaled to 1 million and generalized log-transformed.

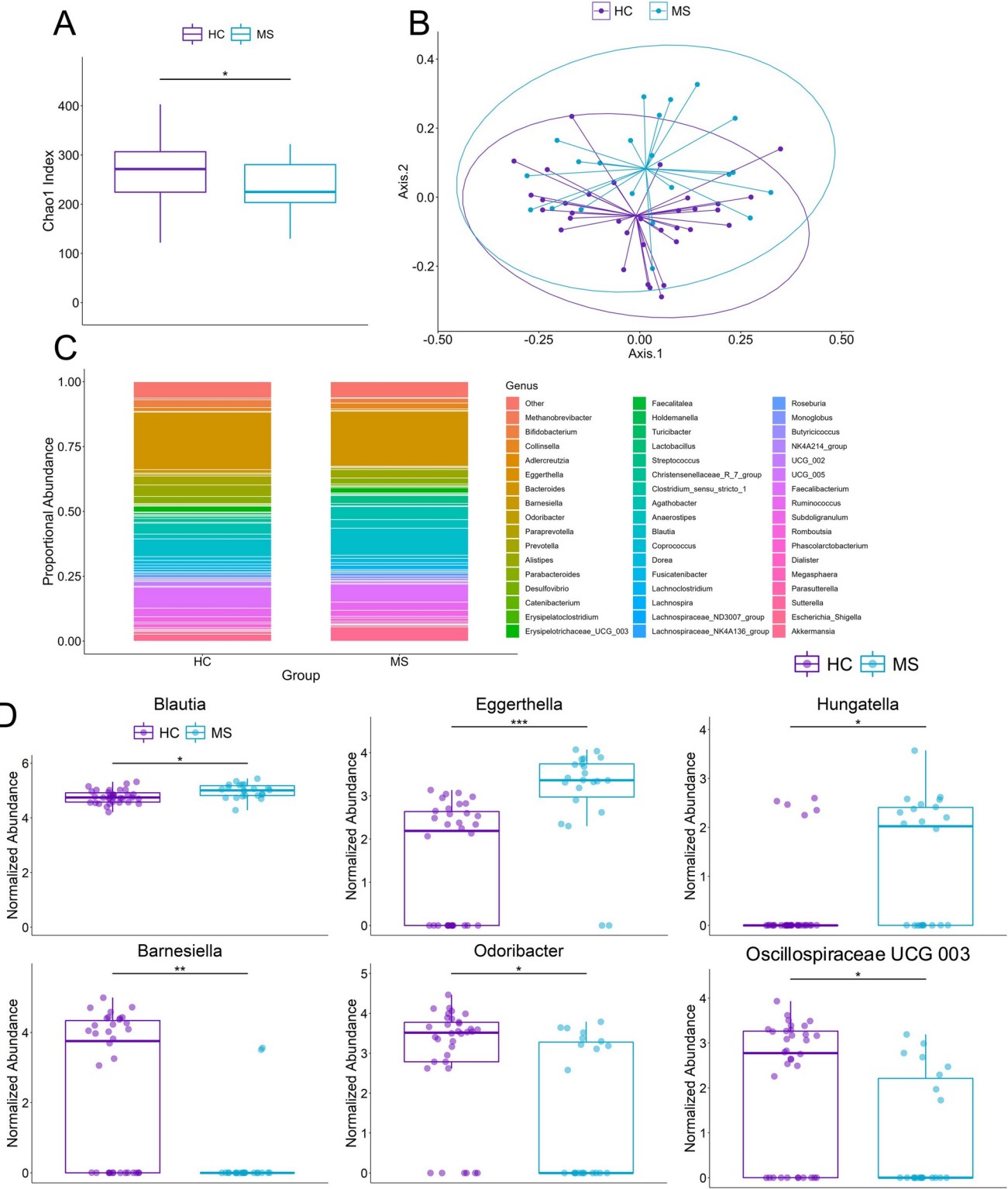

**Fig 4. Gut bacterial microbiome of RRMS patients is different from healthy.** (A) Bacterial ASV richness estimated by Chao1 index showing decreased bacterial richness in MS (p = 0.020). (B) Principal coordinate analysis of Bray-Curtis dissimilarity of HC and MS showing that the microbiome of HC and MS are distinct (PERMANOVA: p = 0.004). Ellipses are visual and do not correspond to any statistical analysis. (C) Relative abundance of top 50 abundant bacteria at the genus level. (D) Differentially abundant bacteria between HC and MS using Wilcoxon signed rank test and adjusted for multiple comparisons with the Benjamini-Hochberg method at a significance level of 0.05. Abundance values are sum-scaled to 1 million and generalized log-transformed. The *, **, and *** symbol indicates p-values of $<0.05$, $<0.01$, $<0.001$, respectively.

reach statistical significance (padj = 0.73; Fig 4C). Overall, the shift in the bacterial microbiome in MS patients in this study reveals a number of genera, several of which have already been described in MS microbiome literature.

### Correlation between gut mycobiome and microbiome

Comparing mycobiome changes with alterations in the bacterial microbiome of the samples show that with decreased bacterial richness comes increased fungal richness. The ITS2/16S ratio is increased in the MS group compared to the HC group (p = 0.040; Fig 5A). Across all samples, bacterial and fungal richness were negatively correlated (R = -0.28, p = 0.042; Fig 5B).

Spearman correlation analysis reveals distinct fungal and bacterial interactions in MS and HC (Fig 5C). The microbial interactions in the HC group are dominated by negative correlations between several bacteria and six fungi. Meanwhile, the MS group exhibited more diverse relationships between the two kingdoms, with numerous positive associations appearing between various fungi and bacteria. MS patients with increased *Blautia* and *Lactococcus* in their stool had decreased *Saccharomyces* and increased *Candida*. *Candida* was also positively associated with *Alistipes* and *Akkermansia*.

Additionally, *Epicoccum* and *Cladosporium* are negatively correlated with *Prevotella*. *Penicillium* is negatively correlated with *Parasutterella* and *Holdemania* and positive correlated with *Terrisporobacter* and *Turicibacter*. *Aspergillus* shares only positive correlations with multiple bacteria: *Paludicola*, an uncultured genus of the *Oscillospiraceae* family, and an uncultured genus of the *Butyricicoccaceae* family. This demonstrates a shift towards potentially more complex bacterial-fungal relationships in MS patients compared to HC.

### Discussion

Our study characterizes the gut fungal microbiome (mycobiome) profile of RRMS patients compared with healthy individuals and identify connections between the mycobiome and bacterial microbiome. Overall, the results of this study demonstrate that the mycobiota of MS patients differs from that of healthy individuals with enrichment of *Candida* and *Epicoccum*, lower relative abundance of *Saccharomyces*, and increased ITS2/16S ratio in MS patients. Thus, our study highlights the important role of the understudied fungal microbiome in MS.

The gut fungal microbiome (mycobiome) profile of individuals with RRMS is a very young field. While this manuscript was under review, Shah et al characterized the gut fungal microbiome (mycobiome) profile of pwMS patients using ITS1 amplicon sequencing and showed that pwMS exhibit gut fungal dysbiosis compared to healthy individuals [18]. They found that *Saccharomyces* and Aspergillus were relatively more abundant in patients with either RRMS, PPMS, SPMS, and CIS compared to healthy individuals [18]. Our study focuses on RRMS patients and adds to the analysis by Shah et al by demonstrating differences in the mycobiome of MS patients compared to healthy individuals using ITS2 amplicon sequencing. We also identified connections between the gut mycobiome and microbiome and examined possible differences in fungal functional profiles. We chose the ITS2 region for fungal profiling instead of ITS1 as a previous study comparing these two regions in fungal identification of the human gut microbiome demonstrated that ITS2 allowed for improved identification of taxa in the gut microbiome samples [30]. Indeed, while Shah et al identified 59 genera, another study in patients with colonic polyps identified 479 genera using the ITS2 region [31]. Additionally, the mycobiota of MS patients differs from that of healthy individuals with enrichment of *Candida*–one of the major fungal genus in the human gut [32]. However, the previous study by Shah et al reported very low levels (<1%) of Candida. We also observed that the *Epicoccum* genus was relatively increased in MS patients while *Saccharomyces* was relatively decreased.

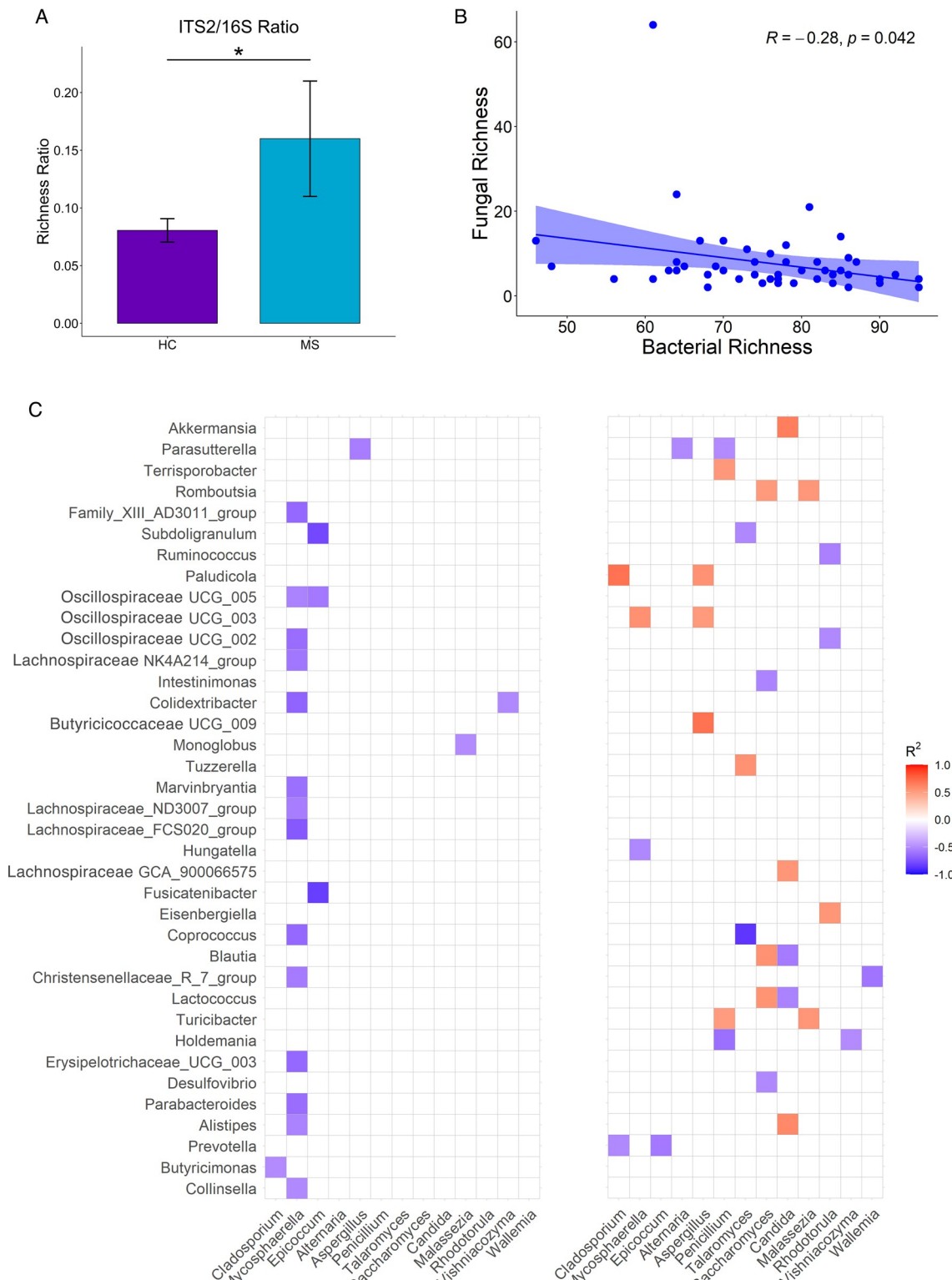

**Fig 5. Correlation between gut mycobiome and microbiome.** (A) Ratio of ITS2 to 16S compared between groups. (B) Linear regression of fungal and bacterial richness shows a negative correlation (Spearman's R = -0.28, p = 0.042). (C) Correlation matrix between bacteria and fungi using Spearman correlation. Values range from -1 to 1 with positive values as orange and red and negative values as purple and blue ($-1 \le R \le 1$). Only statistically significant correlations (p <0.05) are shown.

Additionally, an increased ITS2/16S ratio in MS patients pointed to an overall relative increase in fungal diversity compared to bacterial diversity in the disease state.

We observed that *Ascomycota* and *Basidiomycota* were the main phyla present in MS and HC. Ascomycota and Basidiomycota were negatively correlated with each other which had been previously shown in other autoimmune diseases such as IBD [17] and AS [16]. MS patients had increased *Basidiomycota* compared to *Ascomycota* and an increase in the *Basidiomycota/Ascomycota* ratio, similar to a previously reported ratio in IBD [17]. Meanwhile in AS patients, *Ascomycota* abundance was increased and *Basidiomycota* abundance was decreased [17]. These related findings suggest that shifts in *Ascomycota* and *Basidiomycota* proportions might play a role in the pathogenesis of inflammatory diseases including MS. We detected two fungal genera, *Candida* and *Epiccocum*, with higher relative abundance, and one genus, *Saccharomyces*, with lower abundance in MS compared to HC. Although fungi belonging to the *Candida* species are considered commensal microbes, this genus has also been linked with pathogenicity, especially in individuals with compromised immunity due to antibiotic usage, immunodeficiency, or gut barrier breach [33]. Interestingly, *Candida* species overgrowth had been reported in other autoimmune diseases such as IBD [17,34]. Thus, it is possible that bacterial dysbiosis and inflammatory conditions in the gut create favorable environments for expansion of *Candida* species which can then exacerbate inflammatory diseases such as MS and IBD through induction of pro-inflammatory responses. In contrast to *Candida*, *Saccharomyces*—which is considered a beneficial fungus—has been shown to be decreased in IBD. Our findings showing a decreased abundance of *Saccharomyces* and increased abundance of *Candida* in MS is in concordance with previous findings in IBD [17]. Additionally, in colitis it has been observed that *S. cerevisiae* may exhibit regulatory effects on the host, notably by inducing higher IL-10 production from dendritic cells compared to *Candida* [17]. Moreover, colonic tissue of mice receiving *S. cerevisiae* [35] showed increased expression of IL-10, suggesting that either *S. cerevisiae* has anti-inflammatory potential or it is poorly adapted to an inflammatory environment [17]. As MS is an inflammatory disease, gut mucosal inflammation might explain the decrease in *Saccharomyces* and increase in *Candida*.

Gut bacterial profiling in this study revealed an increased relative abundance of *Hungatella*, *Eggerthella*, and *Blauti*a as well as decreased relative abundance of *Odoribacter*, *Barnesiella*, and an unidentified genus of *Oscillospiraceae*. These findings align with previous MS microbiome studies showing increased relative abundance of *Blautia* and *Eggerthella* in MS [2,3] and decreased relative abundance of *Barnesiella* [1]. Additionally, one study using a mouse model had shown increased *Odoribacter* in mice with relapsing-remitting EAE [36]. Previous studies have shown depletion of *Prevotella* and higher abundance of *Akkermansia* in MS patients [1,2,37], and while we also observed similar trends, these did not reach statistical significance. A number of factors could have contributed to differences in the microbiome profiles of our study, including different 16S rRNA variable region-specific primers, a recently developed analysis pipeline using ASV instead of OTU-based taxonomic mapping, as well as the geographical origin of the samples [38–40]. Our patients are on different DMT treatments which can also modulate gut microbiome and intestinal permeability, but our results did not demonstrate a significant impact of treatment with dimethyl fumarate on the fungal microbiome. Shah et al also did not find a different mycobiome between patients treated with DMTs ($n = 11$) compared to those who were not-treated ($n = 9$) at six months. They also compared the mycobiome changes in three pwMS on dimethyl fumarate (DMF) but found no consistent changes in genus abundances nor alpha diversity after treatment at the six-month point [18]. Additionally, our female to male ratio of 5:1 in this study is higher than 3:1 [1,2] and 2:1 [8] in previous studies, which could contribute to differences. Though our findings combined with those by Shah et al were unable to identify an effect of DMT on the mycobiome, future studies

with larger sample sizes will be required to determine precise relationship between DMT and the human gut mycobiome.

In a healthy state, there is a symbiosis between bacteria, fungi, viruses, and the host. Similar to bacteria, commensal fungi have been shown to play a significant role in maintaining immune homeostasis, and perturbation of the healthy mycobiome can influence the local and peripheral immune systems directly or indirectly through modulation of bacterial populations [41–43]. Fungi may directly interact with the immune system in the gut and influence innate and systemic immune response. Intestinal fungi and their metabolic products may also leak out of the enteric luminal surfaces and activate immune cells [12,44–46]. Besides the direct effect of fungi on the host, relationships between fungi and bacteria might play a critical role in host immunity. Fungi and bacteria can compete for nutrients and adhesion sites and modulate the environment [47,48]. Antibiotics have been linked to increasing growth of fungi in the human intestinal tract [49]. In the mouse model, increased fungal abundance was observed after antibiotic treatment which reduced after antibiotic cessation, pointing towards a balancing mechanism between microbiota and mycobiota in the gut [17,19]. Decreased bacterial diversity favors the growth of certain fungal species, and certain fungi can switch from commensal to pathogenic phenotypes [48]. In this study, we observed decreased bacterial alpha diversity and increased fungal to bacterial richness in MS compared to HC, as evident from the ITS2/16S ratio, suggesting that in MS the gut might favor fungi at the expense of bacteria.

The microbial interactions in the HC group are dominated by negative correlations between several bacteria and fungi. However, our MS group exhibited more diverse relationships between the bacteria and fungi. Specifically, MS patients with increased *Candida* also had increased *Alistipes*, *Akkermansia*, and *Lachnospiraceae GCA*, and those with increased *Saccharomyces* also had increased *Lactococcus*, *Blautia*, and *Romboutsia*. It has been shown that in healthy mucosa, yeast (e.g., *Candida albicans*) is kept at low levels or excluded by the indigenous bacterial microbiota by the mechanism of colonization resistance which is not yet defined [50]. In fact, a previous study showed that germ-free mice are highly susceptible to *Candida* infection and antibiotic treatment leads to an expansion of the fungal population [19,51]. Additionally, studies in IBD and ankylosing spondylitis have shown correlations between the relative abundance of *Saccharomyces* and numerous bacterial genera that have been associated with these diseases. Thus, our study along with previous studies highlight a role of fungal-bacterial interactions in MS as well as other autoimmune diseases.

Functional profiling of the fungal communities in this study suggests modulation of fungal carbohydrate and amino acid pathways. More specifically, while the functional profile of the MS group mycobiome suggested decreased cellulose metabolism by the gut mycobiome (decreased cellobiohydrolase and endoglucanase), starch metabolism by amylase was suggested to be increased. Invertase, which was decreased in MS, can be produced by fungi, mainly *Saccharomyces cerevisiae* [52]. It hydrolyzes sucrose into glucose and fructose and also has antibacterial and antioxidant properties [53]. Sucrose provides a primary substrate for the generation of starch and cellulose [54]. Both cellulose and starch are carbohydrates that can be metabolized into short-chain fatty acids (SCFAs) in the human gut [55]. However, while the metabolism of starches has been shown to increase fecal butyrate levels [56], metabolism of cellulose-related structures [57] leads to increased gut propionate levels [58].

Currently, there are limited tools for functional analysis of the mycobiome, and this study utilized a still growing database of fungal functional profiles. These profiles have been curated by compiling the results of numerous observational and experimental studies spanning over 10,000 fungal genera as of December 2020 [25]. Furthermore, studies examining the functional traits of fungi have shown that many of these traits are conserved at the genus and species levels, thus identification at these levels allows for reasonable functional inferences [59]. However,

functional profiling of fungi is difficult due to the difficulty of culturing fungi and their highly varied, environmentally dependent trophic modes [25]. The quality of these profiles derived from ITS1/2 sequencing data will continue to improve as the field progresses and fungal databases grow. Alternatively, while more cost and time-intensive, whole genome sequencing of the mycobiome would provide detailed data for more reliable functional analysis of the mycobiome.

We acknowledge that our small sample size is a limitation of the study. We didn't observe any effect of different drug treatments including dimethyl fumarate on fungal composition. However due to small sample size, we cannot rule out the impact of treatment on the mycobiome and gut barrier integrity. Another limitation of this study was the different BMI distributions in the HC and MS groups with latter showing higher BMI. Interestingly, BMI was not significantly correlated to most of the differentially abundant bacteria or fungi except for *Eggerthella*. Nonetheless, future studies would benefit from larger sample sizes in order to better isolate the microbiota differences related to MS from those related to BMI and different treatment groups.

## Conclusion

We observed that RRMS patients exhibit an altered gut mycobiome along with an altered gut microbiome. RRMS patients also showed variations in bacterial and fungal relationships with increased ratios of fungal/bacterial abundance and richness. Shifts in fungal functional profiles were also observed in RRMS patients compared to HC. Thus, our data suggest that the mycobiome may play an important role in the pathobiology of MS. Whether changes in the mycobiome play a role in initiating MS pathophysiology or modulating its presentation, or whether the pathology of MS leads to changes in the fungal mycobiome remains an important question for future studies.

## Supporting information

**S1 Fig. Effect of treatment on gut fungal composition.** Principal coordinate analysis of beta diversity of the untreated and treated MS groups using Bray-Curtis dissimilarity. (A) No significant effect of treatment on fungal gut composition was demonstrated when comparing MS without treatment to MS treated with dimethyl fumarate (p = 0.872). (B) No significant effect of treatment on fungal gut composition was demonstrated when comparing untreated MS to MS with either dimethyl fumarate, ocrelizumab, or glatiramer acetate (p = 0.163).
(TIF)

**S2 Fig. Proportional abundance of bacteria at the phylum level.** Stacked bar plots representing the proportional abundance of the top 5 bacterial phyla in MS and HC groups.
(TIF)

**S3 Fig. Relative abundance of differentially abundant families and Genera in HC and MS.** Bar plot showing relative abundances of differentially abundant taxa (p < 0.05) at the family and genus level.
(TIF)

**S1 Table. Metadata for bacterial and fungal sequences uploaded to PRJNA732670.**
(DOCX)

## Author Contributions

**Conceptualization:** Meeta Yadav, Ashutosh K. Mangalam.

**Data curation:** Meeta Yadav, Shailesh K. Shahi, Samantha N. Jensen, Jemmie Hoang, Samuel Cassidy, Heena Olalde, Natalya Guseva, Mishelle Paullus, Catherine Cherwin, Tracey Cho, John Kamholz.

**Formal analysis:** Meeta Yadav, Soham Ali, Rachel L. Shrode, Kai Wang.

**Funding acquisition:** Ashutosh K. Mangalam.

**Investigation:** Meeta Yadav, Soham Ali.

**Methodology:** Meeta Yadav.

**Project administration:** Meeta Yadav, John Kamholz, Ashutosh K. Mangalam.

**Resources:** Tracey Cho, John Kamholz, Ashutosh K. Mangalam.

**Software:** Soham Ali, Rachel L. Shrode, Natalya Guseva.

**Supervision:** Ashutosh K. Mangalam.

**Validation:** Ashutosh K. Mangalam.

**Visualization:** Meeta Yadav, Ashutosh K. Mangalam.

**Writing – original draft:** Meeta Yadav, Soham Ali.

**Writing – review & editing:** Meeta Yadav, Soham Ali, Rachel L. Shrode, Shailesh K. Shahi, Samantha N. Jensen, Jemmie Hoang, Samuel Cassidy, Heena Olalde, Natalya Guseva, Mishelle Paullus, Catherine Cherwin, Kai Wang, Tracey Cho, John Kamholz, Ashutosh K. Mangalam.

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
