## [Decision Letter · Decision Letter 0]

13 Sep 2021

PONE-D-21-26282Multiple Sclerosis Patients have an Altered Gut Mycobiome and Increased Fungal to Bacterial RichnessPLOS ONE

Dear Dr. Mangalam,

Thank you for submitting your manuscript to PLOS ONE. After careful consideration, we feel that it has merit but does not fully meet PLOS ONE’s publication criteria as it currently stands. Therefore, we invite you to submit a revised version of the manuscript that addresses the points raised during the review process.

We look forward to receiving your revised manuscript.

Kind regards,

Christopher Staley, Ph.D.

Academic Editor

PLOS ONE

Journal Requirements:

"AM is one of the inventors of a technology claiming the use of Prevotella histicola to treat autoimmune diseases. AM received royalties from Mayo Clinic (paid by Evelo Biosciences). MY, SA, RS, SS, SJ, JH, SC, HO, NG, MP, CC, KW, TC, JK declare no commercial or financial relationships that could be a potential conflict of interest."

We note that you received funding from a commercial source: Mayo Clinic 

Additional Editor Comments:

This study was of interest as the mycobiome is rarely explored relative to bacterial communities. The reviewers note the small sample size and recommend additional clarifications and analyses to improve the study.

Reviewers' comments:

Reviewer's Responses to Questions

**Comments to the Author**

1. Is the manuscript technically sound, and do the data support the conclusions?

Reviewer #1: Yes

Reviewer #2: Yes

2. Has the statistical analysis been performed appropriately and rigorously? 

Reviewer #1: Yes

Reviewer #2: Yes

3. Have the authors made all data underlying the findings in their manuscript fully available?

Reviewer #1: Yes

Reviewer #2: Yes

4. Is the manuscript presented in an intelligible fashion and written in standard English?

Reviewer #1: Yes

Reviewer #2: Yes

5. Review Comments to the Author

Reviewer #1: This original article is good, in a general typical trend-again, athors are looking for differences in the microbiota in the feces of patients with MS compared to healthy donors. This time, the focus on mushrooms is a special attention to "mycobiota". A number of differences are found (for example, an increase in Candida, Epicoccum, and a decrease in Saccharomyces in MS), but it is unclear what these differences affect and what they are caused by.

Of interest-the microbiota of patients on different therapies is compared (a total of 22 patients with RRRS and 33 healthy donors). Including 6 patients on dimethyl fumarate. The authors write that because of COVID, they have a small group, and therefore there is no comparison between different therapies.

But I would ask them to add a comparison specifically of patients on dimethyl fumarate with other patients, as well as with healthy donors. Dimethylfumart is just known for its fungicidal activity, so it is interesting to see if there is any influence here. For statistical significance, this is not much and must be presented in details

Reviewer #2: The study by Yadav et al., presents data on the gut mycobiome and its relation to bacteria in relapsing MS patients. This manuscript is important, because mycobiota roles in autoimmune mediated diseases like MS are largely unknown. This manuscript is not the first to probe the relationship of fungi and MS, however it does add data points to a still incredibly young field. Enthusiasm for the manuscript is tapered by the small sample size which likely contributes to numerous confounds such as BMI differences, a larger than normal female:male ratio, and 4 DMT populations. Analyses of the mycobiome are scientifically sound, however, there is always inherent risk of garnering misleading data when using 16S or ITS2 data sets to infer species level or functional significance.

That said, I recommend for publication barring changes listed below.

Major comments:

1. This is not the first paper to investigate mycobiome in MS patient’s vs HCs as stated in lines 301-303. Shah et al., 2021, ebiomedicine needs to be cited and discussed in this manuscript.

2. Line 334-335 in discussion should be further fleshed out to include points about DMT influence on gut microbiome and gut barrier integrity.

3. Significant difference in BMI in HC vs MS population needs to be discussed as a limitation of this manuscript.

4. Authors should measure the overall effect of disease status, DMT, BMI and other factors such as ethnicity on the mycobiome and can use the ADONIS test to accomplish this.

5. Discussion about correlative/presumptive nature of inferring species and function of fungi from this data set needs to be included. Line 404-406 starts to do this, however, is not exhaustive enough to provide perspective for the data present in this manuscript.

6. Supplemental figure 2 should be included in the paper and analyzed, as the phyla level abundance 16S analyses are not very exciting compared to previous data in the literature.

Minor comments:

1. The IRB # should be included.

2. This study identified more than double the fungal genera compared to other studies available, authors should elaborate on this in discussion.

3. Paragraph on 387 needs to be more thoroughly cited.

4. Discussion paragraph beginning on line 369 is not focused and does not add much new to the discussion.

5. Final paragraph of discussion does not need to include discussions of COVID impacting patient recruitment.

6. Line 365/366 uses two different fonts.

6. PLOS authors have the option to publish the peer review history of their article (what does this mean?). If published, this will include your full peer review and any attached files.

Reviewer #1: No

Reviewer #2: **Yes: **Howard Weiner

---

## [Author Response · Author response to Decision Letter 0]

27 Oct 2021

We have addressed all the criticisms raised by the reviewers (as detailed in response to reviewer file) and editors and changes are marked by track change in red color with deleted portion strike-out and additions underlined. 

Editor's comments

We have moved funding information to separate section. 

We have also clarified that no funding was received from Mayo Clinic for the study and royalties paid by Mayo Clinic is to an individual (AM) directly for the patent. 

Rebuttal letter and point by point reply to reviewers' comment

Point by Point Reply to Reviewers’ comment

Reviewer #1: This original article is good, in a general typical trend-again, authors are looking for differences in the microbiota in the feces of patients with MS compared to healthy donors. This time, the focus on mushrooms is a special attention to "mycobiota". A number of differences are found (for example, an increase in Candida, Epicoccum, and a decrease in Saccharomyces in MS), but it is unclear what these differences affect and what they are caused by.

Of interest-the microbiota of patients on different therapies is compared (a total of 22 patients with RRRS and 33 healthy donors). Including 6 patients on dimethyl fumarate. The authors write that because of COVID, they have a small group, and therefore there is no comparison between different therapies.

#1 But I would ask them to add a comparison speciﬁcally of patients on dimethyl fumarate with other patients, as well as with healthy donors. Dimethylfumart is just known for its fungicidal activity, so it is interesting to see if there is any inﬂuence here. For statistical signiﬁcance, this is not much and must be presented in details.

 We agree with reviewer and analyzed effect of DMTs including dimethyl fumarate on mycobiome and data is presented in Supplemental figure 1. We also added the same in results and discussed in discussion as highlighted below-

Results-Line 170-174

 “Interestingly, beta-diversity analysis of the MS group did not demonstrate a significant effect of treatment with dimethyl fumarate (p = 0.872) on the mycobiome composition (Supp 1A). Including other disease modifying treatments (DMTs) with at least 3 samples, no significant effect was shown for treatment (p = 0.163; Supp 1B), gender (p = 0.197), or BMI status (p = 0.414).”

Discussion -Line 384-396

“Our patients are on different DMT treatments which can also modulate gut microbiome and intestinal permeability, but our results did not demonstrate a significant impact of treatment with dimethyl fumarate on the fungal microbiome. Shah et al also did not find a different mycobiome between patients treated with DMTs (n=11) compared to those who were not-treated (n=9) at six months. They also compared the mycobiome changes in three pwMS on dimethyl fumarate (DMF) but found no consistent changes in genus abundances nor alpha diversity after treatment at the six-month point [18]. Additionally, our female to male ratio of 5:1 in this study is higher than 3:1 [1, 2] and 2:1 [8] in previous studies, which could contribute to differences. Though our findings combined with those by Shah et al were unable to identify an effect of DMT on the mycobiome, future studies with larger sample size will be required to determine precise relationship between DMT and the human gut mycobiome.”

Reviewer #2: The study by Yadav et al., presents data on the gut mycobiome and its relation to bacteria in relapsing MS patients. This manuscript is important, because mycobiota roles in autoimmune mediated diseases like MS are largely unknown.

This manuscript is not the ﬁrst to probe the relationship of fungi and MS, however it does add data points to a still incredibly young ﬁeld. Enthusiasm for the manuscript is tapered by the small sample size which likely contributes to numerous confounds such as BMI differences, a larger than normal female:male ratio, and 4 DMT populations. Analyses of the mycobiome are scientiﬁcally sound, however, there is always inherent risk of garnering misleading data when using 16S or ITS2 data sets to infer species level or functional signiﬁcance.

That said, I recommend for publication barring changes listed below. 

 Major comments:

1. This is not the first paper to investigate mycobiome in MS patient’s vs HCs as stated in lines 301-303. Shah et al., 2021, ebiomedicine needs to be cited and discussed in this manuscript.

Thanks for bringing it to our attention. We have removed that sentence saying our is the first study. We have cited and discussed paper by Shah et al on fungal microbiome published in EBiomedicine. Starting discussion paragraphs now reads-

“Our study characterizes the gut fungal microbiome (mycobiome) profile of RRMS patients compared with healthy individuals and identify connections between the mycobiome and bacterial microbiome. Overall, the results of this study demonstrate that the mycobiota of MS patients differs from that of healthy individuals with enrichment of Candida, and Epicoccum, with lower relative abundance of Saccharomyces, and increased ITS2/16S ratio in MS patients. Thus, our study highlights the important role of the understudied fungal microbiome in MS.

The gut fungal microbiome (mycobiome) profile of individuals with RRMS is a very young field. While this manuscript was under review, Shah et al characterized the gut fungal microbiome (mycobiome) profile of pwMS patients using ITS1 amplicon sequencing and showed that pwMS have fungal dysbiosis compared to healthy individuals [18]. They found that Saccharomyces and Aspergillus were relatively less abundant in group of patients with either RRMS, PPMS, SPMS, and CIS compared to healthy individuals [18]. Our study focuses on RRMS patients and adds to the analysis by Shah et al by demonstrating differences in the mycobiome of MS patients compared to healthy individuals using ITS2 amplicon sequencing. We also identified connections between the gut mycobiome and microbiome and examined possible differences in fungal functional profiles.”

2. Line 334-335 in discussion should be further fleshed out to include points about DMT influence on gut microbiome and gut barrier integrity.

We have further addressed how DMTs may influence the gut microbiome and gut barrier integrity. Now the discussion (lines 384-396).

“Our patients are on different DMT treatments which can also modulate gut microbiome and intestinal permeability, but our results did not demonstrate a significant impact of treatment with dimethyl fumarate on the fungal microbiome. Shah et al also did not find a different mycobiome between patients treated with DMTs (n=11) compared to those who were not-treated (n=9) at six months. They also compared the mycobiome changes in three pwMS on dimethyl fumarate (DMF) but found no consistent changes in genus abundances nor alpha diversity after treatment at the six-month point [18] Additionally, our female to male ratio of 5:1 in this study is higher than 3:1 [1, 2] and 2:1 [8] in previous studies, which could contribute to differences. Thus, our findings combined with Shah et al suggests that DMT don’t affect the mycobiome, however future studies with larger sample size will be required to determine precise effect of DMT on mycobiome.”

3. Significant difference in BMI in HC vs MS population needs to be discussed as a limitation of this manuscript.

We have added how BMI can be a limitation to this study on lines 459-461

“Another limitation of this study was the different Body Mass Index (BMI) distributions in the HC and MS groups with latter showing higher BMI. Interestingly, BMI was not significantly correlated to most of the differentially abundant bacteria or fungi except for Eggerthella.”

4. Authors should measure the overall effect of disease status, DMT, BMI and other factors such as ethnicity on the mycobiome and can use the ADONIS test to accomplish this.

As per your suggestion we have added supplemental figure showing these analysis as mentioned in #2 and #3. Since no patients in the HC group were taking any DMTs, treatment and disease are codependent variables, so we performed the PERMANOVA analysis with adonis2 on just the MS group to measure the overall effects of treatment, gender, and BMI status. We separately analyzed the overall effects of disease status, gender, and BMI status on both the HC and MS groups and have revised the results section on lines 171-176 to reflect all these results.

“Principal coordinate analysis of beta diversity using Bray-Curtis dissimilarity (Fig 1B) demonstrated distinct clustering of the mycobiome of MS compared with HC (p = 0.011), while gender (p = 0.32) and BMI status (p = 0.73) did not show a significant effect on mycobiome composition. Interestingly, beta-diversity analysis of the MS group did not demonstrate a significant effect of treatment with dimethyl fumarate (p = 0.872) on the mycobiome composition (Supp 1A). Including other disease modifying treatments (DMTs) with at least 3 samples, no significant effect was shown for treatment (p = 0.163; Supp 1B), gender (p = 0.197), or BMI status (p = 0.414).”

5. Discussion about correlative/presumptive nature of inferring species and function of fungi from this data set needs to be included. Line 404-406 starts to do this, however, is not exhaustive enough to provide perspective for the data present in this manuscript.

We have further discussed the presumptive nature of inferring fungi species and function from our data set on lines 444-453:

“Currently, there are limited tools for functional analysis of the mycobiome, and this study utilized a still growing database of fungal functional profiles. These profiles have been curated by compiling the results of numerous observational and experimental studies spanning over 10000 fungal genera as of December 2020 (FungalTraits paper). Furthermore, studies examining the functional traits of fungi have shown that many of these traits are conserved at the genus and species levels, thus identification at these levels allows for reasonable functional inferences (31763752). However, functional profiling of fungi is difficult due to the difficulty of culturing fungi and their highly varied, environmentally dependent trophic modes24. The quality of these profiles derived from ITS1/2 sequencing data will continue to improve as the field progresses and these databases grow.” 

6. Supplemental figure 2 should be included in the paper and analyzed, as the phyla level abundance 16S analyses are not very exciting compared to previous data in the literature.

We have moved the previous Supplemental figure 2 to Figure 4C and have moved the previous Figure 4C to the supplement. We have also provided an analysis of the relative abundances of the most common genera. We decided to try to capture 95% of the bacteria with the genera depicted in this figure, so we showed the top 50 genera to do this.

 Minor comments:

1. The IRB # should be included.

We have included the IRB number on line 76 (201512717).

2. This study identified more than double the fungal genera compared to other studies available, authors should elaborate on this in discussion.

As per your suggestion we have added this to our discussion

“The use of ITS2 region for fungal profiling instead of ITS1 allowed for improved identification of taxa in the gut microbiome samples, as previously comparing these two regions in fungal identification of the human gut microbiome demonstrated (PMID:30283425). Indeed, while Shah et al identified 59 genera, a similar study in patients with colonic polyps that profiled the ITS2 region instead identified 479 genera (PMID:28821976).”

3. Paragraph on 387 needs to be more thoroughly cited.

We have cited appropriate references in the paragraph as advised by the reviewer and paragraph read as below-

“Invertase, which was decreased in MS, can be produced by fungi, mainly Saccharomyces cerevisiae [52]. It hydrolyzes sucrose into glucose and fructose and also has antibacterial and antioxidant properties [53]. Sucrose provides a primary substrate for the generation of starch and cellulose [54]. Both cellulose and starch are carbohydrates that can be metabolized into short-chain fatty acids (SCFAs) in the human gut [55]. However, while the metabolism of starches has been shown to increase fecal butyrate levels [56], metabolism of cellulose-related structures [57] leads to increased gut propionate levels [58]”

4. Discussion paragraph beginning on line 369 is not focused and does not add much new to the discussion.

We have made this paragraph more succinct and highlight the point that, in addition to the potential role of fungi and bacteria separately in disease, the correlations suggest that interactions between these two kingdoms may also be playing a role. We had not made this clear in our previous version and hope this paragraph now better serves its purpose.

“The microbial interactions in the HC group are dominated by negative correlations between several bacteria and fungi. However, our MS group exhibited more diverse relationships between the bacteria and fungi. Specifically, MS patients with increased Candida also had increased Alistipes, Akkermansia, and Lachnospiraceae GCA, and those with increased Saccharomyces also had increased Lactococcus, Blautia, and Romboutsia. It has been shown that in healthy mucosa, yeast (e.g., Candida albicans) is kept at low levels or excluded by the indigenous bacterial microbiota by the mechanism of colonization resistance which is not yet defined [50]. In fact, a previous study showed that germ-free mice are highly susceptible to Candida infection and antibiotics treatment leads to an expansion of fungal population [19, 51]. Additionally, studies in IBD and ankylosing spondylitis have shown correlations between the relative abundance of Saccharomyces and numerous bacterial genera that have been associated with these diseases. Thus, our study along with previous studies highlight a role of fungal-bacterial interactions in MS as well as other autoimmune diseases.”

5. Final paragraph of discussion does not need to include discussions of COVID impacting patient recruitment.

We have removed our statement about COVID impacting patient recruitment as suggested by the reviewer.

6. Line 365/366 uses two different fonts.

We have made the fonts the same between the lines pointed out by the reviewer.

---

## [Decision Letter · Decision Letter 1]

14 Feb 2022

Multiple sclerosis patients have an altered gut mycobiome and increased fungal to bacterial richness

PONE-D-21-26282R1

Dear Dr. Mangalam,

We’re pleased to inform you that your manuscript has been judged scientifically suitable for publication and will be formally accepted for publication once it meets all outstanding technical requirements.

Kind regards,

Christopher Staley, Ph.D.

Academic Editor

PLOS ONE

Additional Editor Comments (optional):

Reviewers' comments:

Reviewer's Responses to Questions

**Comments to the Author**

1. If the authors have adequately addressed your comments raised in a previous round of review and you feel that this manuscript is now acceptable for publication, you may indicate that here to bypass the “Comments to the Author” section, enter your conflict of interest statement in the “Confidential to Editor” section, and submit your "Accept" recommendation.

Reviewer #1: All comments have been addressed

Reviewer #2: All comments have been addressed

2. Is the manuscript technically sound, and do the data support the conclusions?

Reviewer #1: Yes

Reviewer #2: Yes

3. Has the statistical analysis been performed appropriately and rigorously? 

Reviewer #1: Yes

Reviewer #2: Yes

4. Have the authors made all data underlying the findings in their manuscript fully available?

Reviewer #1: Yes

Reviewer #2: Yes

5. Is the manuscript presented in an intelligible fashion and written in standard English?

Reviewer #1: Yes

Reviewer #2: Yes

6. Review Comments to the Author

Reviewer #1: Authors made all recommended corrections and could be accepted to the journal. All comments and explanations are understandable.

Reviewer #2: The authors have addressed all my comments. The paper has addresses an important question related to the microbiome in MS.

7. PLOS authors have the option to publish the peer review history of their article (what does this mean?). If published, this will include your full peer review and any attached files.

Reviewer #1: No

Reviewer #2: **Yes: **Howard L Weiner

---

## [Editor Report · Acceptance letter]

18 Apr 2022

PONE-D-21-26282R1 

Multiple sclerosis patients have an altered gut mycobiome and increased fungal to bacterial richness 

Dear Dr. Mangalam:

I'm pleased to inform you that your manuscript has been deemed suitable for publication in PLOS ONE. Congratulations! Your manuscript is now with our production department. 

Kind regards, 

on behalf of

Dr. PLOS Manuscript Reassignment 

Staff Editor

PLOS ONE